# Evaluation of Crude and Recombinant Antigens of *Schistosoma japonicum* for the Detection of *Schistosoma mekongi* Human Infection

**DOI:** 10.3390/diagnostics13020184

**Published:** 2023-01-04

**Authors:** Jose Ma. M. Angeles, Atcharaphan Wanlop, Minh-Anh Dang-Trinh, Masashi Kirinoki, Shin-ichiro Kawazu, Aya Yajima

**Affiliations:** 1Department of Parasitology, College of Public Health, University of the Philippines Manila, Manila 1000, Philippines; 2National Research Center for Protozoan Diseases, Obihiro University of Agriculture and Veterinary Medicine, Obihiro 080-8555, Japan; 3Institute of Malariology, Parasitology and Entomology Ho Chi Minh, Ho Chi Minh 700000, Vietnam; 4Laboratory of Tropical Medicine and Parasitology, Dokkyo Medical University, Tochigi 321-0293, Japan; 5World Health Organization Regional Office for South-East Asia, New Delhi 110002, India

**Keywords:** *Schistosoma mekongi*, *Schistosoma japonicum*, crude antigen, recombinant antigen, ELISA

## Abstract

Asian schistosomiasis caused by the blood fluke *Schistosoma mekongi* is endemic in northern Cambodia and Southern Lao People’s Democratic Republic. The disease is mainly diagnosed by stool microscopy. However, serodiagnosis such as enzyme-linked immunosorbent assay (ELISA) with soluble egg antigen (SEA), has been shown to have better sensitivity compared to the stool examination, especially in the settings with a low intensity of infection. To date, no recombinant antigen has been assessed using ELISA for the detection of *S. mekongi* infection, due to the lack of genome information for this schistosome species. Thus, the objective of this study is to evaluate several recombinant *S. japonicum* antigens that have been developed in our laboratory for the detection of *S. mekongi* infection. The crude antigen SjSEA and recombinant antigens Sj7TR, SjPCS, SjPRx-4, and SjChi-3 were evaluated in ELISA using serum samples positive for *S. mekongi* infection. The cross-reaction was checked using sera positive for *Ophistorchis viverrini*. ELISA results showed that *S. japonicum* SEA at low concentrations showed better diagnostic performance than the recombinant antigens tested using the archived serum samples from Cambodia. However, further optimization of the recombinant antigens should be conducted in future studies to improve their diagnostic performance for *S. mekongi* detection.

## 1. Introduction

Schistosomiasis is a parasitic disease affecting 78 countries worldwide, with approximately 229 million people requiring preventive treatment in 2018 [1]. This includes Asian schistosomiasis caused by the blood fluke *Schistosoma mekongi,* which affects communities in the Mekong River Basin in southern Lao People’s Democratic Republic (Lao PDR) and northern Cambodia [2]. Control strategies for schistosomiasis should include accurate, reliable, and inexpensive diagnostic methods that will monitor infection dynamics and treatment efficacy. However, technological gaps in the current diagnostic methods used by the endemic countries for *S. mekongi* infection impose significant limitations on the epidemiological analysis and elimination programs.

Current parasitological methods such as Kato–Katz have been producing unreliable results, varying from one day to the next for the same patient. This significantly underestimates the infection levels, particularly in low transmission settings such as in Cambodia and Lao PDR. As compared with fecal microscopic examination, serology provides a sensitive tool for the diagnosis of schistosomiasis, especially for low-intensity infections [3,4,5,6]. A sensitive and specific serological test might therefore be useful in assessing human infections in the endemic areas. This test will also be useful in confirming the lack of transmission in areas where disease elimination has been achieved. Serodiagnosis using sodium metaperiodate SMP-ELISA has shown high sensitivity and specificity for *S. mekongi* [7]. However, the production of SMP-ELISA is challenged by the unstable supply of *S. mekongi* eggs in the pre-elimination setting, and therefore, alternative serodiagnostic tools with similarly high sensitivity and specificity should also be explored.

The availability of the whole genome sequences of the *Schistosoma* species, including *Schistosoma mansoni*, *Schistosoma haematobium,* and *Schistosoma japonicum* has led to the production of recombinant proteins used in different studies on drug discovery, diagnostics, and vaccine development. However, this is not the case for *S. mekongi*. The lack of the publicly available genome of *S. mekongi* has been a hurdle in developing accurate serodiagnostic tools for this parasitic disease in the endemic areas.

Several schistosome-specific antigens have been evaluated for their diagnostic potential in *Schistosoma japonicum* infection. Table 1 shows the recombinant antigens evaluated in this study for the diagnosis of *S. japonicum* infection in human samples, with their sensitivities and specificities in previous studies. These recombinant antigens have been tested using a serum from individuals positive for S. japonicum, as confirmed through microscopic analysis of their feces. One of these antigens, *S. japonicum* peroxiredoxin-4 (SjPrx-4), was analyzed in our previous study, with results suggesting that this enzyme may play a role as an antioxidant in *S. japonicum* to deal with oxidative stress [8]. Serological analysis using human samples of rSjPrx-4 showed 83.3% sensitivity and 86.7% specificity. When rSjPrx-4 was combined with the *S. japonicum* thioredoxin peroxidase-1 (rSjTPx-1) antigen, the sensitivity improved to 90.0% [8]. Another enzyme, *S. japonicum* phytochelatin synthase (SjPCS). SjPCS is an enzyme that catalyzes the biosynthesis of phytochelatin, which is capable of scavenging and detoxifying heavy metals [9]. In *Schistosoma mansoni*, it was seen to be expressed in the eggs, schistosomula, and adult stages [10]. SjPCS was evaluated for the detection of human schistosomiasis, with sensitivity and specificity results of 73.3% and 83.3%, respectively [11].

Tandem repeat proteins (TRPs) are often targets of humoral responses for helminthic parasites [12]. One of the tandem repeats we have evaluated in our previous study is Sj7TR, which showed an 80% sensitivity and 93.3% specificity [13]. We have also evaluated its immunolocalization in different life stages of *S. japonicum,* and our results showed that it was expressed in the eggs, schistosomules, and juvenile adults [14]. On the other hand, rSjChi3 is a multiepitope protein constructed by combining the epitope sequences of SjSAP4 (saposin), SjTPx-1, Sj23LHD (a large hydrophilic protein), and SjSAP5 (unpublished). Serological evaluation for human schistosomiasis of this chimeric protein showed an 90% sensitivity and 93.3% specificity.

Although these are antigens found in *S. japonicum*, they also might have the potential to be useful in detecting *S. mekongi* infections, since these two species don’t share epidemiological foci in Asia. *S. japonicum* is endemic in China, the Philippines, and some parts of Indonesia, while *S. mekongi* is endemic in the areas along the Mekong River belonging to Cambodia and Lao P.D.R. This study, therefore, aims to evaluate these recombinant *S. japonicum* antigens for the detection of *S. mekongi* human infections as compared to the crude SjSEA using ELISA.

## 2. Materials and Methods

### 2.1. Antigens

#### 2.1.1. Soluble Egg Antigen (SjSEA)

SjSEA was prepared following standard procedures with some modifications [15]. The intestines of *S. japonicum*-infected mice were digested with 0.02% Pronase E (Actinase E in PBS). This was twice homogenized at 10,000 rpm for 1 min, and incubated at 37 °C for 2 h with agitation. The homogenate was then filtered through a steel mesh and centrifuged at 13,000 rpm for 5 min at 4 °C. The residue was then dissolved in PBS, mixed, and centrifuged again. 10× volume of 0.05% collagenase solution was added to the resulting pellet. The suspension was incubated at 37 °C for 1 ½ h with agitation, and then washed twice with PBS and centrifuged at 11,000 rpm for 2 min at 4 °C. Filtration was conducted to the solution to collect the schistosome eggs. The eggs were suspended in cold, distilled water and then lyophilized. The lyophilized egg was homogenized in carbonate buffer solution and stored for 2 days at 4 °C with constant stirring. The homogenate was centrifuged at 14,000 rpm for 1 h with the resulting supernatant being filtered through a 0.45 μm syringe filter. The antigen was kept at −80 °C until use.

#### 2.1.2. Recombinant Antigens

Preparation of recombinant antigens from *S. japonicum* was conducted as previously described [13]. In brief, pET28 vector containing the inserted gene was transfected into Escherichia coli BL21 cultured in SOB medium. Recovery of the recombinant proteins was performed using Ni-NTA agarose, which was then dialyzed and eluted with 20 mM Tris, pH 8.0. Evaluation of the integrity and purity of the proteins was conducted using 15% polyacrylamide gel electrophoresis (SDS-PAGE) under denaturing conditions and subsequent Coomassie Brilliant Blue staining. The concentration of each expressed protein was determined using BCA protein assay.

### 2.2. Serum Samples

Archived serum samples collected from Cambodia were used in the evaluation of SjSEA and the recombinant antigens. This includes negative samples collected in the non-endemic area of Phnom Penh (*n* = 31) and *S. mekongi* egg-confirmed serum samples from an endemic area in Cambodia (*n* = 28). To check cross-reaction, 21 *Opisthorchis viverrini* positive serum samples collected in Thailand were also evaluated. In addition, 15 positive and 16 negative samples of *S. japonicum* from the Philippines, confirmed through stool microscopy, were also tested for the cell-free expressed recombinant proteins. A panel of serum samples (*n* = 31) purchased from BioreclamationIVT (Baltimore, MD, USA) was used to calculate the cut-off values as mean + 3SD.

### 2.3. Enzyme-Linked Immunosorbent Assay (ELISA)

ELISA was conducted as previously described in our published study [13]. In brief, horseradish peroxidase (HRP)-conjugated anti-human IgG goat serum was used for the secondary antibody, and 3,3′,5,5′-tetramethylbenzidine was used as the substrate for HRP. The wells of the microplates were sensitized separately with the following antigen concentrations per well: SjSEA at 20 μg/mL and 2 μg/mL; Sj7TR, SjPCS, and SjPrx-4 at 200 ng/mL; and SjChi3 at 200 ng/mL and 20 ng/mL. As for cell-free expressed proteins, 200 ng/mL of GST-tagged rSjTPx-1 and 100 and 200 ng/mL of HIS-tagged rSjTPx-1 were used for each well. Proteins were diluted with carbonate/bicarbonate buffer at pH 9.6. After blocking with 1% bovine serum albumin (BSA) in phosphate-buffered saline with 0.05% Tween 20 (T-PBS) (T-PBS-1%BSA), the antigen-coated well was filled with the serum. 0.1 mL of the test serum was diluted 1:400 with T-PBS-1%BSA and 0.1 mL of the secondary antibody was diluted 1:10,000. Optical density (OD) was measured at 450 nm using a microplate reader. All the tests were conducted in triplicates.

### 2.4. Statistical Analysis

Diagnostic sensitivity and specificity of the recombinant antigens were calculated for the crude SjSEA and the recombinant antigens. The sensitivity, specificity, and predictive values were calculated for the crude and recombinant antigens through the online software MedCalc (https://www.medcalc.org/calc/diagnostic_test.php (accessed on 24 July 2021)).

## 3. Results

The ELISA results showed that SjSEA and the recombinant antigens have high OD values with most of the *S. mekongi* positive samples (Figure 1). However, non-endemic samples from Cambodia were also giving high OD values with the recombinant antigens. Among the recombinant antigens, the highest number of positive results with the non-endemic negative samples were shown by SjChi3 at a 200 ng/mL concentration, and the lowest with Sj7TR. The results from the recombinant antigen were surpassed by SjSEA at 2 μg/mL concentration. Cross-reaction with *O. viverrini* was seen in SjSEA at 20 μg/mL with 16 samples. This has greatly improved with 2 μg/mL SjSEA, showing no cross-reaction. Among the recombinant proteins, only SjChi3 showed no cross-reaction with *O. viverrini*. In previous studies, these antigens have already been evaluated against other parasitic infections such as soil-transmitted helminthiases, malaria, amoebiasis, and paragonimiasis, where they have shown no or very minimal cross-reactions [8,11,13].

Table 2 shows the sensitivity, specificity, and predictive values calculated for each antigen. SjSEA at a 2 μg/mL concentration has the highest diagnostic activity for the detection of *S. mekongi* with 96.4% sensitivity, 93.5% specificity, 93.1% positive predictive value, and 96.7% negative predictive value.

## 4. Discussion

Mass drug administration in endemic areas including Cambodia and Lao PDR has resulted in a reduction of its prevalence to less than 5%, based on stool microscopy [16]. However, an underestimation of the true prevalence of schistosomiasis in areas with low endemicity is usually seen when the diagnosis is completed using the Kato–Katz technique [17]. Therefore, a more sensitive and specific diagnostic test is needed when targeting disease elimination.

The use of recombinant proteins has been proven useful in the diagnosis of *S. japonicum* in humans [13] and animals [18,19,20]. However, the results of this study have shown otherwise. The seropositivity seen in negative samples from Cambodia and positive samples for *O. viverrini* suggests that the recombinant antigens in their present form could not be used in the region endemic for both *S. mekongi* and *O. viverrini*. Therefore, the development of improved antigens either from *S. japonicum* or *S. mekongi* should still be conducted.

Although tedious, the production of soluble egg antigens from *S. japonicum* is well-established, due to the availability of the laboratory set-up for both animal and snail infections in maintaining the parasite’s life cycle. Unfortunately, this laboratory infection system is not yet available for *S. mekongi*. Therefore, the results of this study, showing that SjSEA is useful in the diagnosis of *S. mekongi* infection, could take advantage of the laboratory production of SjSEA. Massive production of SjSEA could be conducted to meet the possible needs for surveillance in endemic countries.

Serological tests have been found useful in schistosomiasis-endemic areas with a low intensity of infection as compared to Kato–Katz stool examinations. Therefore, the lack of a sensitive serological test for the detection of *S. mekongi* infection will underestimate the true prevalence of the disease and might hinder its elimination in endemic foci. The results of this study on the possible use of SjSEA at a low concentration of 2 μg/mL for *S. mekongi* diagnosis will improve the serodiagnostic capabilities that will impact the control strategies conducted in Cambodia and Lao PDR.

## Figures and Tables

**Figure 1 diagnostics-13-00184-f001:**
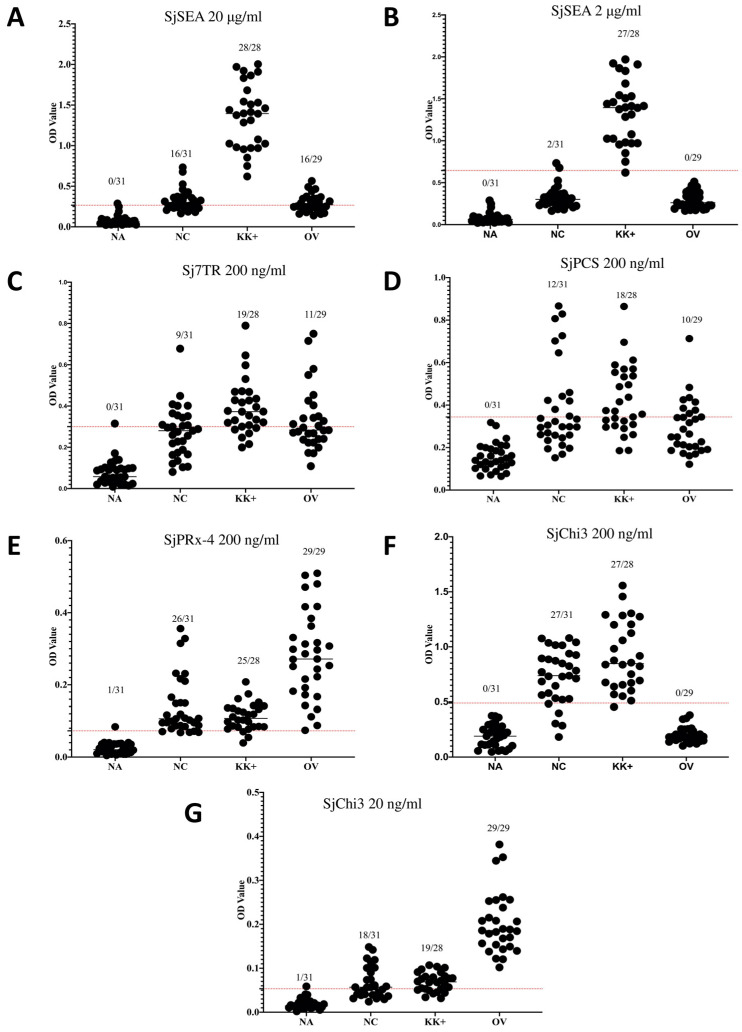
ELISA results of crude SjSEA and the recombinant antigens. The graph shows the ELISA results using crude antigen SjSEA at (**A**) 20 μg/mL and (**B**) 2 μg/mL concentrations, recombinant antigens (**C**) Sj7TR, (**D**) SjPCS and (**E**) SjPRx-4 at 200 ng/mL and multiepitope recombinant protein SjChi3 at (**F**) 200 ng/mL and (**G**) 20 ng/mL. Legend: NA, serum samples from non-endemic Americans used for the calculation of the cut-off values; NC, serum samples from the non-endemic area in Cambodia; KK+, serum samples from endemic areas in Cambodia confirmed as *S. mekongi* egg positive through Kato–Katz technique; OV, serum samples from Thailand confirmed positive for *O. viverrini* using stool microscopy. Solid lines represent mean values; dotted lines represent the cut-off values.

**Table 1 diagnostics-13-00184-t001:** Diagnostic potentials of the recombinant antigens in the detection of *S. japonicum* human infection.

Abbreviation	Sj Recombinant Proteins	Sensitivity	Specificity	Reference
rSj7TR	Tandem repeat protein	80.0%	93.3%	[12]
rSjPCS	Phytochelatin synthase	73.3%	83.3%	[11]
rSjPrx-4	Peroxiredoxin-4	83.3%	86.7%	[8]
rSjChi3	Chimeric protein consisting of selected peptides from SjSAP4, SjTPx-1, Sj23LHD, and SjSAP5	90%	93.3%	Unpublished

**Table 2 diagnostics-13-00184-t002:** Sensitivity and specificity of *S. japonicum* antigens in the detection of *S. mekongi* infection.

Antigens	Concentration	Sensitivity	Specificity	NPV *	PPV *
SjSEA	20 μg	100%	48.4%	100%	63.6%
SjSEA	2 μg	96.4%	93.5%	96.7%	93.1%
rSj7TR	200 ng	67.9%	71.0%	71.9%	70.4%
rSjPCS	200 ng	82.1%	35.5%	65.5%	60%
rSjPrx-4	200 ng	89.3%	16.1%	62.5%	49%
rSjChi3	200 ng	89.3%	3%	80%	50%
rSjChi3	20 ng	67.9%	71.0%	59.1%	51.4%

* NPV, negative predictive values; PPV, positive predictive values.

## Data Availability

The data relating to this manuscript are available upon request.

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
