# Peer review of "Evaluation of Crude and Recombinant Antigens of Schistosoma japonicum for the Detection of Schistosoma mekongi Human Infection"

_diagnostics, 2023, doi:10.3390/diagnostics13020184_

Round 1

Reviewer 1 Report

Evaluation of crude and recombinant antigens of Schistosoma japonicum for the detection of Schistosoma mekongi human in infection  

Type of manuscript: Communication 

Section: Diagnostics  

General comments 

This work describes ELISA study using Schistosoma japonicum antigens with the serum of infected patients with Schistosoma mekongi . The methodology is not new, but I know that the absence of the S.mekongi genome makes new studies more difficult.  However, I think essential details, such as animal ethics and statistical analysis, are missing.  

Furthermore, the discussion needs to be improved.  

To improve the manuscript, I suggest some changes. 

Introduction 
Line 59: Please, use italic in S. japonicum.  

Line 62: Something is missing in this line. Please, re write it.  

Line 62-65: Need to be re write. It is not clear.  

Line 54-78: I understand that all these antigens come from S. japonicum. Still, please clarify if all of them have been serologically tested in patients infected with this or other species of Schistosoma.  

Material and Methods 

It is imperative to have approval for Laboratory Animal Regulation in the workplace. Also, because the human serum is used for the analysis, approval from the corresponding organism is required.  

Furthermore, please complete the statistical analysis correctly and how you calculate the specificity and sensibility.  

 Line 84: There is a point in antigens that should be deleted.   

Line 86: Please, use italic in S. japonicum.  

Line 85: Is this method original? Please, add a reference.  

Line 99: Please, use italic in S. japonicum. 

Line 110: Please, use italic in S. mekongi.  

Results 

Line 151: Please, use italic in O. viverrini.  

Figure 1: The quality of the figure must be improved.  

Table 2: Please, use italic.  

Discussion 

Line 176-185: Please write again. It is not clear. 

Conclusion 

This could be added to the discussion. I think it is not necessary to have a separate section.

Author Response

Thank you very much for your comments and suggestions to improve the manuscript. Here are our responses:

Introduction 
Line 59: Please, use italic in S. japonicum.  

  • The word has been italicized.

Line 62: Something is missing in this line. Please, re write it.

  • The sentence has been rewritten.

Line 62-65: Need to be re write. It is not clear.

  • The statements have been rewritten.

Line 54-78: I understand that all these antigens come from S. japonicum. Still, please clarify if all of them have been serologically tested in patients infected with this or other species of Schistosoma.

  • This has been clarified by adding the sentence “These recombinant antigens have been tested using serum from individuals positive for S. japonicum as confirmed through microscopic analysis of their feces.” in Lines 65-67.

Material and Methods 

It is imperative to have approval for Laboratory Animal Regulation in the workplace. Also, because the human serum is used for the analysis, approval from the corresponding organism is required.

  • A sentence for the approval of animal use in the production of SjSEA has been added in the Institutional Review Board Statement stating “This study was approved by the Committee on the Ethics of Animal Experiments of Dokkyo Medical University (Permit No. 0006).
  • For human serum, the ethical statement in the Institutional Review Board Statement section has been revised into “The use of archived human samples in this study was approved by the Committee for Research Ethics of Obihiro University of Agriculture and Veterinary Medicine (Approval no. 2020-01-2)”.

Furthermore, please complete the statistical analysis correctly and how you calculate the specificity and sensibility

  • The sentence “The sensitivity, specificity and predictive values were calculated for the crude and re-combinant antigens through the online software MedCalc (https://www.medcalc.org/calc/diagnostic_test.php)” has been added in Lines 155-157. Table 2 has been revised with the inclusion of the positive and negative predictive values..

 Line 84: There is a point in antigens that should be deleted.

  • The point has been deleted.

Line 86: Please, use italic in S. japonicum.

  • The word has been italicized.

Line 85: Is this method original? Please, add a reference.

  • A reference has been added (Boros D, Warren K. Delayed hypersensitivity-type granuloma formation and dermal reaction induced and elicited by a soluble factor isolated from Schistosoma mansoni J. Exp. Med. 1970, 132: 488–507).

Line 99: Please, use italic in S. japonicum. 

  • The word has been italicized.

Line 110: Please, use italic in S. mekongi.

  • The word has been italicized.

Results 

Line 151: Please, use italic in O. viverrini.

  • The word has been italicized.

Figure 1: The quality of the figure must be improved.  

  • Figure 1 was enlarged to improve its quality.

Table 2: Please, use italic.

  • The word has been italicized.

Discussion 

Line 176-185: Please write again. It is not clear. 

  • This has been rewritten as “Although tedious, the production of soluble egg antigen from S. japonicum is well-established due to the availability of laboratory set-up for both animal and snail infection in maintaining the parasite’s life cycle. Unfortunately, this laboratory infection system is not yet available for mekongi. Therefore, the results of this study, showing that SjSEA is useful in the diagnosis of S. mekongi infection, could take advantage of the laboratory production of SjSEA. Massive production of SjSEA could be done to meet the possible needs for surveillance in endemic countries.”

Conclusion 

This could be added to the discussion. I think it is not necessary to have a separate section.

  • The Conclusion has been added to the Discussion part.

Reviewer 2 Report

The authors propose an immunological test for diagnosing Schistosoma mekongi since coprological diagnose is not fully reliable. They use crude extract of S. japonicum or their recombinant fractions. ?

I did not get clearly what were the human infections that coexists in the S. mekongi area. There were apparently also Opistorchis viverrini infections. Was S. japonicum also present?. Could you clarify the epidemiological situation in Laos and Cambodia. Are S. japonicum and S. mekongi really different species, since Schistosoma are prone to make hybrids. If they are different species give some references.

Minor items

l 51 S mekongi: in italics

l151: O viverrini in italics

l24SEASEA5: Mol in italics

In table 1 SEA and in text SjSEA

SjSEA was positive for O viverrini at 20µg or 2 µg?

table 2 S japonicum in italics

Author Response

The authors propose an immunological test for diagnosing Schistosoma mekongi since coprological diagnose is not fully reliable.

  • Thank you very much for your comments and recommendations.

They use crude extract of S. japonicum or their recombinant fractions. ?

  • For this study, we have used the crude extract from the schistosome egg for the production of the soluble egg antigen (SjSEA) and the recombinant antigens Sj7TR, SjPrx-4, SjPCS, and SjChi3 produced through coli expression system.

I did not get clearly what were the human infections that coexists in the S. mekongi area. There were apparently also Opistorchis viverrini infections. Was S. japonicum also present?. Could you clarify the epidemiological situation in Laos and Cambodia. Are S. japonicum and S. mekongi really different species, since Schistosoma are prone to make hybrids. If they are different species give some references.

  • S. japonicum and mekongi do npt coexist in the same epidemiological foci. S. japonicum is endemic in China, the Philippines and some parts of Indonesia while S. mekongi is endemic in the areas along the Mekong River belonging to Cambodia and Lao PDR. We have added this information in Line 90-91.

Minor items

l 51 S mekongi: in italics

  • The word has been italicized.

l151: O viverrini in italics

  • The word has been italicized.

l24SEASEA5: Mol in italics

  • The word has been italicized.

In table 1 SEA and in text SjSEA

  • Ths has been revised accordingly.

SjSEA was positive for O viverrini at 20µg or 2 µg?

  • This has been changed to 20 µg.

table 2 S japonicum in italics

  • The word has been italicized.

Round 2

Reviewer 1 Report

Dear Authors, 

Thank you for answering all my suggestions. I think the Ms has improved, and I hope all the scientific audience can enjoy it. 

Best wishes. 

Reviewer 2 Report

Problems are now mended. It can be published as is.